# Influence of the Chemical Properties of Cereal Grains on the Structure and Metabolism of the Bacteriome of *Rhyzopertha dominica* (F.) and Its Development: A Cause–Effect Analysis

**DOI:** 10.3390/ijms251810130

**Published:** 2024-09-20

**Authors:** Olga Kosewska, Sebastian Wojciech Przemieniecki, Mariusz Nietupski

**Affiliations:** Department of Entomology, Phytopathology and Molecular Diagnostics, Faculty of Agriculture and Forestry, University of Warmia and Mazury in Olsztyn, Prawocheńskiego 17, 10-720 Olsztyn, Poland; mariusz.nietupski@uwm.edu.pl

**Keywords:** storage pests, *Rhyzopertha dominica*, microbiome, food chemistry, wheat, barley

## Abstract

*Rhyzopertha dominica* causes significant economic losses in stored cereals. Insects’ digestive tract microbiome is crucial for their development, metabolism, resistance, and digestion. This work aimed to test whether the different chemical properties of different wheat and barley grain cultivars cause disturbances in insect foraging and rearrangements of the structure of the *R. dominica* microbiome. The results indicated that grain cultivars significantly influence the microbiome, metabolism, and insect foraging. Most observed traits and microbiome structures were not correlated at the species level, as confirmed by ANOSIM (*p* = 0.441). However, the PLS-PM analysis revealed significant patterns within barley cultivars. The study found associations between C18:2 fatty acids, entomopathogenic bacteria, an impaired nitrogen cycle, lysine production of bacterial origin, and insect feeding. The antioxidant effects also showed trends towards impacting the microbiome and insect development. The findings suggest that manipulating grain chemical properties (increasing C18:2 and antioxidant levels) can influence the *R. dominica* microbiome, disrupting their foraging behaviours and adaptation to storage environments. This research supports the potential for breeding resistant cereals, offering an effective pest control strategy and reducing pesticide use in food production.

## 1. Introduction

The species *Rhyzopertha dominica* (F.) (Coleoptera: Bostrichidae), also known as the lesser grain borer, was first recorded under the bark of *Quercus suber* (cork oak) and *Cytisus spinosus* (spiny broom) as early as 1849. It is believed that *R. dominica* is native to India, and most early reports are from this location.Today, this insect has a cosmopolitan range and has adapted to a diet consisting of cereal grains and other stored products [1]. Thus, *R. dominica* has become one of the most dangerous storage pests worldwide. It occupies a leading position among insects, causing damage to grain storage areas, and its presence can lead to significant economic losses in agriculture and the food industry [2]. Grain losses during storage due to insect feeding account for 10 to 30% in developing countries, while in developed countries, they are 5–10% [3,4]. Assuming that, on average, these losses are 20% of the stored grain worldwide, approximately 420 million tonnes of grain are lost annually during storage [5]. *R. dominica* causes a large proportion of these losses. This insect is particularly threatening in countries with warm climates, such as Australia, India, and Ethiopia [4,5,6]. In recent years, however, it has been observed that this species can adapt rapidly to unfavourable abiotic conditions and has significantly changed its habitat preferences, making it thrive in temperate climate zones [7,8].

This pest attacks a wide range of cereals, including wheat, barley, rice, and maize, seriously threatening food security [9]. Its life cycle is short, and females lay large numbers of eggs, contributing to rapid population growth. Its high fecundity and ability to survive in a wide range of temperatures and humidities make *R. dominica* an extremely difficult-to-control storage pest [10]. A key factor influencing the development and foraging of *R. dominica* is its microbiome, which is the collection of microorganisms inhabiting the insect’s body. The microbiome plays an important role in insect function, influencing digestion, resistance to pathogens, and reproductive capacity [11,12]. It is increasingly recognised that interactions between a host and its microbiome can shape host fitness and evolutionary potential [13,14], increasing tolerance to abiotic stress, enabling food digestion, or providing protection against pathogens [15,16]. For example, a reduction in the diversity of the microbiome in the gut of tadpoles can reduce host fitness and tolerance to thermal stress in a matter of days [17], and the acquisition of pesticide-degrading bacteria can immediately confer pesticide resistance to bugs [18]. Symbiotic bacteria found in the digestive tract of insects can help, among other things, to digest difficult-to-digest nutrients such as cellulose [19]. These microorganisms can protect insects from pathogens and produce essential vitamins and amino acids [20]. Studies show that the diet is one of the factors directly influencing the microbiome’s composition, which can affect the insect’s health and development [21]. A change in the diet can lead to a change in the composition of the microbiome and affect the metabolic functions of the bacteria inhabiting the digestive tract of insects, as has been observed in *Bombus terrestris* (ground bumblebee), among others [22]. Similarly, studies by Montagna et al. [20] on the red palm weevil (*Rhynchophorus ferrugineus*) showed differences in bacterial composition depending on whether the insects were fed on palm tissues or an apple-based substrate.

Studies of the diet of the *R. dominica* show that different types of food can affect the development and health of this insect [23]. A protein-rich diet can accelerate larval development and increase larval survival [24]. Also, other chemical components of the food, such as carbohydrates, antioxidants, or the fat content, can affect the development of storage pests [25], including *R. dominica*. The importance of the microbiome for the health and development of *R. dominica* is increasingly appreciated. Understanding the interactions between the diet, microbiome, and physiology of *R. dominica* may contribute to developing new control strategies for this pest. However, there are few scientific articles on the *R. dominica* microbiome [21] and how it changes with different diets. Evidence of symbiotic interactions with the microbiota among some insects indicates the need for significant research on the susceptibility of the pest to a loss of the gut microbiota. This may lead to new perspectives on control strategies for *R. dominica* in grain storage facilities. In particular, one of the most important problems related to the control of *R. dominica* is that it becomes resistant to the active substances used for its control, that is, deltamethrin and phosphate hydrogen [26,27]. Therefore, a strategic solution is to reduce agrophages through plant varietal advances [7]. Current research is increasingly focusing on exploiting the natural resistance of different plant species, which can significantly reduce the need for synthetic pesticides and improve productivity, profitability, and environmental compliance in crop production [9]. This study aimed to investigate the relationship between the microbiome of *R. dominica* and its development and foraging. Particular attention was paid to the effects of the diet on the composition of the microbiome and the physiology of the insect and to test how the different chemical properties of different cultivars of common wheat and barley grains cause disturbances in the foraging of *R. dominica* and changes in the structure of the microbiome of this pest. This study aimed to provide new information that can be used to develop more effective methods to control *R. dominica* populations and minimise losses in grain stores.

The study assessed the natural resistance of different wheat and barley species to *R. dominica* feeding. The hypothesis adopted is that the grains of the cereal species and cultivars studied show different levels of natural resistance to *R. dominica* attack related to their different chemical compositions. The verification of the adopted research hypothesis was carried out by confirming the influence of the chemical properties of the tested wheat and barley cultivars on the composition and activity of the symbiotic microbiome of *R. dominica* and confirming the close correlation between the process of food digestion by *R. dominica* and the enzymatic activity and interaction with the symbiotic microbiome of this insect. The chemical properties of the grains also affect the composition and activity of the insect’s symbiotic microbiome, which may modify the digestive processes of these organisms. It was assumed that the food digestion process in *R. dominica* is based on enzymatic activity and an interaction with the symbiotic microbiome of this insect.

## 2. Results

### 2.1. Relationship between the Development of R. dominica and Chemical Parameters of Cereal Grains

Significant differences between the wheat cultivars tested were observed for traits such as consumption, number of imago, and dry matter (SM), ash (Ash), total protein (TP), crude fibre (CFibre) and water-soluble sugar (WSC) contents. The consumption intensity differed between the Varius cultivar, where the highest grain weight loss was recorded (4.6 g), and the Impresja cultivar, which had the lowest weight loss (0.41 g). Analogous results were observed for the number of imago. All cultivars tested differed highly (*p* < 0.001) in the dry matter content. The highest dry matter content was observed in the grain of the Impresja cultivar, a medium content in the Rusałka cultivar, and the lowest content in the grain of the Varius cultivar. For the ash (Ash) and crude fibre (CFibre) contents, the Varius cultivar had significantly higher contents than the other cultivars. Each variant differed highly significantly in the total protein (TP) content. The highest TP content characterised the Varius cultivar, the Rusałka cultivar had an average content, and the Impresja cultivar had the lowest content. The Rusałka cultivar differed significantly and had a lower crude fat content than the other variants (CFat). The content of water-soluble sugars differed highly significantly between each variant. The highest WSC content was characterised by the grains of the Rusałka cultivar and the lowest by the Impresja cultivar. The antioxidant content did not differ significantly among cultivars; however, the Varius cultivar had the highest antioxidant potential and the Impresja cultivar had the lowest, with a *p*-value of 0.98 (trend). The NMDS graph showed that the Impresja and Rusałka variant samples had low dissimilarity (Figure 1).

No significant difference was observed in insect foraging for the barley cultivars tested. The grain mass loss measurements were as follows: 2.56 g for cultivar Ismena, 2.0 g for cultivar Radek, and 1.41 g for cultivar Trofeum. However, grain chemical parameters differed significantly in total protein (TP), crude fat (CFat) and water-soluble sugar (WSC) contents. The protein content was significantly higher in the grain of the Trofeum cultivar than in the other variants. On the other hand, the crude fat content was significantly higher in the grain of the Ismena cultivar, with the lowest fat content observed in the Trofeum cultivar. The WSC content was significantly lower in the grain of the Radek cultivar than in the other variants. No significant differences were observed regarding antioxidant activity, only a trend (*p* = 0.086) between the Ismena cultivar with the highest antioxidant activity and the Trofeum cultivar with the lowest antioxidant activity. Based on the NMDS results, partial dissimilarity was observed between the Trofeum and Radek cultivars. For the Ismena cultivar, unclear relationships were observed due to the high dissimilarity within this variant (Figure 2).

### 2.2. Analysis of the Bacteriobiome of R. dominica

By analysing the percentages of the different bacterial genera identified in the digestive tract of the *R. dominica* feeding on different wheat and barley cultivars (Table 1), we noted significant microbial diversity, which depends on both the type of grain and the generation of the insect.

An analysis of the bacterial diversity in the digestive tract of the *R. dominica* feeding on different wheat and barley cultivars showed an apparent dominance of *Pantoea* and *Ralstonia* bacteria in most cultivars and generations. In the first generation, the wheat cultivar Impresja and the barley cultivar Ismena resulted in the highest microbial diversity. In contrast, in the second generation, the wheat cultivar Impresja and the barley cultivar Radek resulted in the highest microbial diversity. In generation one, the wheat cultivar Impresja was dominated by *Pantoea* (18.9%) and *Staphylococcus* (20.3%), with *Ralstonia* (14.3%) also making a significant contribution. The wheat cultivar Rusałka showed a high proportion of bacteria from the genera *Staphylococcus* (28.7%), *Pantoea* (18.6%) and *Ralstonia* (11.4%). The Varius cultivar was also dominated by bacteria of the genera *Pantoea* (24.2%), *Ralstonia* (27.7%) and *Stenotrophomonas* (11.1%). For barley, the following bacterial genera dominated in generation one of the *R. dominica* fed the Ismena cultivar: *Stenotrophomonas* (25.3%), *Ralstonia* (22.8%), and *Pantoea* (12.3%). The insects fed the Trofeum cultivar showed a high presence of bacteria of the genera *Pantoea* (29.2%), *Staphylococcus* (14.5%) and *Acinetobacter* (12.8%), while the insects fed the Radek cultivar were characterised by a significant dominance of bacteria of the genus *Staphylococcus* (54.9%). In the second insect generation, in the variant fed wheat of the Impresja cultivar, bacteria of the genera *Stenotrophomonas* (22.3%), *Pantoea* (19.7%) and *Ralstonia* (19.9%) were most prevalent. In the variant fed the Rusałka P II cultivar, there were increases in the proportions of bacteria of the genera *Pantoea* (35.3%), *Acinetobacter* (21.1%), and Serratia (17.7%). The insects fed the cultivar Varius were characterised by a dominance of *Pantoea* (36.0%), *Massilia* (12.4%), and *Ralstonia* (11.0%). For barley-fed insects in generation two, the insects fed the Radek cultivar variant were characterised by a significant presence of *Candidatus Sulcia* (19.6%), *Xanthomonas* (14.3%), and *Pantoea* (10.8%). The insects fed the Ismena cultivar showed a dominance of bacteria of the genera *Pseudomonas* (22.6%) and *Ralstonia* (15.6%), while those fed the Trofeum cultivar showed a high proportion of bacteria of the genera *Pantoea* (28.0%) and *Ralstonia* (19.2%).

The analysis of diversity indices for both generations showed significant differences in dominance, diversity, and evenness between insects fed wheat and barley cultivars (Table 1). Based on the diversity results, it was observed that in terms of Simpson’s dominance, generation one insects fed barley cultivars Trofeum and Radek presented higher dominance than other variants from generation one. The Shannon diversity index showed that almost all cultivars used to feed generation one were not different from each other, while they were significantly different from the wheat cultivar Impresja (2.0590ab). The Pielou equality index, on the other hand, was significantly lower in the barley cultivar variants Trofeum and Radek for generation one and higher in the cultivar variants Impresja, Rusałka, Varius and Ismena for generation one. In the second generation, it was observed that in terms of Simpson’s dominance, the wheat cultivar Rusałka had the highest value of the index (0.2579a). In contrast, the barley cultivar Ismena showed the lowest dominance (0.1322b). The Shannon diversity index for the second generation showed that the barley cultivar Ismena had the highest value (2.2490a). The index values for the other cultivars in the second generation were similar and fluctuated between 1.7 and 2.0. The Pielou equality index was highest for the wheat cultivar Impresja (0.6445a). The lowest value of the equality index was recorded for the wheat cultivar Rusałka (0.4831b).

Based on the PERMANOVA results (Figure 3A), the generation and cultivar were found to have highly significant (*p* < 0.01) effects on rearrangements of the structure of the *R. dominica*. The coefficient of determination, R^2^, results were relatively low for these factors and amounted to 0.078 for the generation and 0.193 for cultivars. The R^2^ value for residuals was high, at 0.563.

The Bray–Curtis dissimilarity results for the bacterial OTUs (Figure 3A) revealed the formation of four distinct dissimilarity groups. The first group includes *Berkiella*, *Moraxella*, *Escherishia-Shigella*, *Sporosaricina*, *Conchiformibius*, and *Aureimonas*. The second group includes *Bacillus*, *Alcaligenes*, and *Microvirga*. The third group includes *Pantoea*, *Acinetobacter*, *Serratia*, *Xanthomonas*, *Stenotrophomonas*, *Massilia*, *Ralstonia*, and *Pseudomonas*. The fourth group includes *Corynebacterium*, *Brevundimonas*, *Brevibacterium*, *Staphylococcus*, *Sulcia*, *Kocuria*, *Enterobacterales*, *Lysobacter*, *Xanthobacteraceae*, *Ancylobacter*, *Cupriavidus*, *Comamonadaceae*, *Caulobacter*, *Sphingomonas*, *Sphingobium*, *Paenibacillus*, *Bodea*, *Rhizobium*, *Variovorax*, *Afipia*, *Burkholderia*, and *Methylobacterium*.

Bacteriobiome structural diversity data obtained from Bray–Curtis dissimilarity results for cultivars and generations (Figure 3B) allowed two distinct groups to be distinguished. The first group was formed from seven samples. It included two samples, each of Radek from generations one and two, and one sample each for generation one of the Rusałka, Impresja, and Trofeum cultivars. The microbiomes for the remaining samples were included in the second group in two subgroups. The first subgroup comprised 17 samples, that is, one sample each of the Radek cultivar for generations one and two, two samples each of Rusałka and Impresja for generation one, all samples of Ismena for generation one and Impresja for generation two, two samples of Varius for generation one and one for generation two, and the background and one sample of Trofeum for generation two. The second subgroup of this clade was formed from 13 samples comprising all Ismena and Rusałka samples for generation two, two samples each of Trofeum for generation one and two for generation two, Varius for generation two and one sample of Varius for generation one. The visualisation of dissimilarity in the NMDS plot indicated outlier samples, including the Radek cultivars for generations one and two, one sample of Rusałka for generation one, and Impression and Trofeum for generation one.

Based on the ANOSIM results (Figure 3C), it was observed that for the generation factor, there was moderate variation between the bacteriobiomes. Nevertheless, this differentiating group of OTUs significantly distinguished the bacteriobiomes from each other (*p* < 0.01). Similar results were observed for cultivars. However, the *p*-value was smaller (*p* < 0.05).

### 2.3. Metabolism of the R. dominica Bacteriobiome

The heatmap analysis results for the microbiome metabolism (Figure 4, left side) showed that the bacteriobiome of the Radek cultivar was the most different from the other bacteriobiomes and had the lowest metabolic activity.

Only β-gal parameters and GNT utilisation were highly active. The cultivar Ismena was an intermediate cultivar between Radek and the other cultivars. This cultivar was characterised by high activity of α-chym, N-ace-β-gluko, and α-gal, and the utilisation of MAL, NAG, and ADI. The microbiome of the Impresja cultivar was characterised as the only one with poor tryptophan metabolism. It showed high or very high activity for 17 of 29 traits, and the highest activity was found for phospho_n-AS-BI, URE and β-gal, and the utilisation of CAP, ARA, and GLU. The Varius cultivar had high or very high activity for 23 of the 29 traits tested, and the most active traits were α-gal, aryl cys, α-fuk, β-glu and CAP, ARA, GLU, ADI, CIT, PAC, and MLT utilisation. The Rusałka and Trofeum cultivars had a high similarity, with 14 traits with high activity in common, and aryl cys and PAC had the strongest activities. The bacteriobiomes for these cultivars differed in fos_n-AS-BI, α-gal, FOS, and N-ace-ß-glu activities, and the utilisation of ARA, MAL, and MAN.

Based on the analysis of the predicted multitraits of bacteriobiomes (Figure 4, right side), an analogous grouping of cultivars was observed for the metabolic activity traits. The most different cultivar from the other cultivars was the Radek cultivar. The bacteriobiomes for this cultivar showed high activity for 21 of the 68 traits analysed. These traits included electron-harvesting metabolism involving minerals, the detoxification and utilisation of arsenic, and bacteria inhabiting the digestive tract or human food pathogens. For the bacteriobiomes characteristic of the Ismena cultivar, high activity of traits related to the utilisation of nitrogen compounds for respiration, a trait complex for the degradation of organic compounds, nitrate reduction, mammalian pathogenicity, oxidation of methanol, manganese compounds and dark oxidation of sulphur compounds, methylotrophy, denitrification, degradation of cellulose, urea, xylanose lignin, and invertebrate pathogens were observed. The bacteriobiomes for the Varius cultivar were characterised by N-fixation, chemoheterotrophy, mammalian and plant pathogenic trait complex, sulphur and hydrogen oxidation in the dark, fermentation, utilisation of arsenic and iron compounds, knallgas bacteria, photoautotrophy, and nitrification. The bacteriobiomes for the Rusałka cultivar were characterised by fermentation potential, sulfuretted hydrogen oxidation in the dark, ammonification, the presence of predatory or exoparasitic and mammalian pathogens, nitrate reduction, chitin breakdown, nitrogen fixation, and chemoheterotrophy. The bacteriobiome for the Impresja cultivar was characterised by the possession of traits such as organic matter decomposition, urea decomposition, chitin decomposition, respiration involving nitrogen and its compounds, chemoheterotrophy, the presence of predatory or exoparasitic and mammalian and plant pathogens, nitrogen fixation, chemoheterotrophy, and aerobic anoxygenic phototrophy. The bacteriobiomes for the Trofeum cultivar had characteristics similar to those of Impresja. The differences were the low activity of chitin degradation and respiration with nitrogen and its compounds, and the possession of traits such as human pathogens (pneumonia), nitrite respiration, hydrocarbon degradation, invertebrate parasites, oil bioremediation, fermentation, and ammonification.

### 2.4. Global Analysis of the Relationships between the Tested Parameters

Based on the PLS-PM analysis (Figure 5), including the combined results for wheat and barley, all groups of observed variables were observed to affect insect development significantly. The grain property group was moderately correlated (r = 0.640) with insect development. After analysing the effects of individual observed characteristics (manifest variables) in the grain property group, TP and WSC contents were observed to be negatively correlated with ash, fat, CF, and antioxidant contents. Grain fatty acid contents were also significantly correlated, but moderately, with insect development.

Within this group, negative correlations were observed between C18:2 and C14:0, C15:0, C16:0, C18:3, C20:0, and C20:1. The structure of the bacteriobiome was influenced significantly, but a weak correlation was shown. For bacteriobiome metabolism, a moderate but significant correlation was shown, and multiple bacteriobiome traits were moderately negatively correlated with insect development. A very high correlation was observed between the structure of the bacteriobiome and its metabolism, while it was highly negatively correlated with mulitraits. Nevertheless, the significance between the three groups describing the bacteriobiome showed a trend (*p*-values between 0.1 and 0.05). The results of the PLS-PM analysis constructed for each cereal species (Appendix A) were significant for both wheat and barley. When comparing the graphs, weak correlation coverage was observed between the graphs. The direct effects of grain properties, grain fatty acids and the pathways of the bacteriobiome or its metabolism and traits were different depending on the cereal species. Among the main results, a negative correlation was observed for grain chemical traits in wheat, but a positive correlation of these traits was observed for barley in relation to insect development. For intermediate pathways, the only pathway similarly correlated was grain properties—bacteriobiome traits—insect development, but correlations were high for wheat and low for barley. A large proportion of the OTU variables did not overlap, corresponding to the ANOSIM score for the species. Variables such as dry matter, total protein, and crude fibre contents were correlated within grain properties in both models. For fatty acids, similar within-group correlations were observed for C15:0, C18:0, C18:1c9, and C18:2 in both models. An important observation is that the grain species affected the bacteriobiomes quite differently, but in both cases, the changes were completely correlated with changes in metabolism. Furthermore, the direct effect of grain properties on bacteriobiome metabolism was almost completely negatively correlated in wheat, while it was positively correlated in barley.

As the development parameters were lower for each barley cultivar than the wheat cultivars, it was impossible to establish unequivocally traits negatively correlated with development parameters. Nevertheless, despite the lack of significant differences, in general terms, both Ismena and Radek cultivars tended to result in a lower imago weight and number. As previously observed, these are related to the low TP and C18:2 contents but also to a weaker fermentation potential, metabolism of compounds without light, and a weaker potential to utilise different carbon sources (Figure 6).

In the network analysis, the traits most correlated with the imago number and weight were determined, which largely coincided with the data obtained using PCA. Grain chemical traits strongly correlated with changes in imago number and imago weight, but also indirectly with consumption and the mass of dust, including the TP, WSC and fatty acid 18:2 contents. Bacterial OTUs included *Pantoea*, *Paenibacillus*, *Caulobacter*, *Sphingomonas* and *Massilla*. Among the metabolic traits correlated were urease, NOB, and aryl cys activities, and the utilisation of MNE, GLU, MAN, ARA, and PAC. Of the predicted traits, bacteriobiome traits were observed to be correlated with dark hydrogen oxidation, dark sulfur/sulfite oxidation, fermentation, predatory or exoparasitic, animal parasites or symbionts, human pathogens, chemoheterotrophy, aerobic anoxygenic phototrophy, photoheterotrophy, plant pathogens, and phototrophy. Note that a direct correlation between the mass of imago and the number of imago was only observed with traits such as C18:2 levels, a human pathogen, dark hydrogen oxidation, dark sulphur oxidation, and fermentation (Figure 6).

## 3. Discussion

The microbiome of *R. dominica* still needs to be understood, especially in terms of its role in the development and function of the insect. The bacteria present in the body of *R. dominica* were first described in 1934 [28]. In 2017, Okude et al. [29] identified a unique endosymbiont in *R. dominica*. In 2023, Xue et al. [21] analysed the microbiota composition of *R. dominica* populations from different geographical locations, both wild and laboratory-reared, to understand the influence of the geographical origin on the microbial diversity of these insects.

The results show significant relationships between selected chemical parameters and insect feeding intensity and development (Figure 5). The chemical compositions of the tested wheat and barley grains differ (Figure 1 and Figure 2), and these differences can be both attractant and repellent factors for insects [30]. Our study showed that total protein and water-soluble sugar contents were key parameters affecting insect development and foraging (Figure 5 and Figure 6). Our results suggest a positive correlation between a higher total protein content, increased foraging intensity, and the development of *R. dominica*. The wheat cultivar Varius, with the highest total protein content, also had the highest number and weight of imago, suggesting that a high protein content may favour their development. In contrast, the wheat cultivar Impresja, with the lowest protein content among the cultivars tested, showed the lowest foraging intensity (Figure 1). These observations are compatible with previous studies, which also indicated a relationship between the protein content and the foraging activity of storage pests. Perišić et al. [24] evaluated the feeding preference and reproduction of *R. dominica* on different cereals, highlighting that triticale, with its higher protein content, influenced insect development better than other cereals. Cinco-Moroyoqui et al. [31], on the other hand, showed that higher protein intake correlates with increased reproductive success. Mariey et al. [32] confirmed that the foraging intensity of *R. dominica* on different barley cultivars is correlated with protein and carbohydrate contents, which is consistent with the previously cited studies.

Similar relationships were observed for water-soluble sugar (WSC) contents. A higher WSC content was positively correlated with beetle foraging intensity (Figure 5). Soluble sugars are an easily digestible energy source that can positively affect insect development. The WSC content can affect the amount of available sugars in the grains, affecting their attractiveness as food for storage pests. Therefore, studies on the relationship between the WSC content and the amylose/amylopectin ratio in different cereal species can provide crucial information on the food preference and development of storage pests [25].

The crude fat content differed significantly between the wheat and barley cultivars tested. The results indicate that the cultivar Rusałka had the lowest crude fat content, which may have influenced the lower intensity of insect feeding on this cultivar (Figure 1). Crude fats may be less attractive to insects than proteins and sugars, which explains the observed relationships. It is, therefore, worth focusing more on the contents of individual fatty acids and their effects on insects. The fatty acid analysis noted that the lower contents of most fatty acids were associated with better development of *R. dominica*. In contrast to linoleic acid (C18:2), the lower contents of saturated and polyunsaturated fatty acids may indicate less complex lipid structures, making it easier for insects to absorb energy and nutrients from the grain. A link was observed between the linoleic acid content of grains, a group of potentially entomopathogenic bacteria, and insect foraging (Figure 6). The high C18:2 content of the grains may also have affected the rearrangement of the microbiome, which may interfere with insect function and reduce their foraging efficiency. Nietupski et al. [33], in their study on the development of *S. granarius*, suggest that the development intensity of this species is significantly correlated with higher contents of unsaturated and saturated fatty acids in the grain. A study by Kordan et al. [34] also confirms that the fatty acid content influences the essential functions of *S. granarius*; in particular, fatty acids such as C18:1 and C18:2 have impacts on the survival of this pest. Similar relationships were observed in a study by Kosewska et al. [35] of *S. oryzae*, which developed significantly better on wheat cultivars that had a higher C18:2 fatty acid content.

Furthermore, an association has been observed between linoleic acid (C18:2), a group of potentially entomopathogenic bacteria, and insect feeding behaviour. This supports the hypothesis that a diet low in LA, a fatty acid with inhibitory effects on undesirable microbiota development, influences the rearrangement of the microbiome, thereby impairing insect function. This relationship is not strictly dependent on the insect species, as many insects show a high correlation between the LA content and the development of an undesirable microbiota or internal infection [36,37]. The poor overlap of correlations between the PLS-PM plots constructed for the individual cereal tannins (Appendix A) indicates the different effects of some of the grain traits studied on the direct and indirect foraging of *R. dominica*. However, it should be noted that, despite the different correlations between changes within a group of traits, there were often extremely different correlations between groups.

The analysis of the diversity of the bacterial microbiome of *R. dominica*, based on the presented dendrogram (Figure 3), indicates the existence of four distinct groups of bacteria. Each group may reflect different microbiological ecosystems or ecological niches associated with *R. dominica*. The microflora of insects is shaped by several factors, which, in addition to diet, include the geographical location, environmental conditions (temperature, humidity, and climate), host species, developmental stage, or integration with other microorganisms [21]. Many metabolites produced by bacteria are influenced by environmental changes, including insect diet changes [36].

It was observed that four groups of bacteria formed, resulting from the influence of the examined cereal cultivar factors. Group 1 consists of *Berkiella*, *Moraxella*, *Escherichia-Shigella*, *Sporosarcina*, *Conchiformibius*, and *Aureimonas*. These bacteria are often associated with animals’ digestive systems and aquatic and soil environments. For example, *Escherichia-Shigella* is mainly linked to the intestines of animals. The presence of this group may suggest that *R. dominica* inhabits environments rich in organic matter, where it may come into contact with soil or water, or that the insect has a complex digestive system supporting these bacteria. Group 2 includes *Bacillus*, *Alcaligenes*, and *Microvirga*. *Bacillus* species are soil bacteria often involved in the degradation of organic matter. *Alcaligenes* and *Microvirga* are also associated with soil and water environments. This group’s presence might indicate that soil ecosystems are a key environment for *R. dominica*. These bacteria may aid the insect in digesting organic matter and detoxification processes. Group 3 comprises *Acinetobacter*, *Serratia*, *Xanthomonas*, *Stenotrophomonas*, *Massilia*, *Ralstonia*, *Pantoea*, and *Pseudomonas*. These bacteria are widespread in aquatic and soil environments, as well as in the digestive systems of animals. Notably, *Pseudomonas* and *Ralstonia* are known for their ability to degrade complex organic compounds. This group suggests strong associations with organic matter decomposition and possibly pathogenicity, as some of these bacteria can be plant or animal pathogens. *Pantoea* spp., especially *P. agglomerans*, is a bacterium transmitted from plant food to an insect. It has a wide environmental adaptation, and so it is a facultative insect symbiont, but its function is not fully understood. Group 4 is the most diverse, containing *Corynebacterium*, *Brevundimonas*, *Brevibacterium*, *Staphylococcus*, *Sulcia*, *Kocuria*, *Enterobacterales*, *Lysobacter*, *Xanthobacteraceae*, *Ancylobacter*, *Cupriavidus*, *Comamonadaceae*, *Caulobacter*, *Sphingomonas*, *Sphingobium*, *Paenibacillus*, *Bodea*, *Rhizobium*, *Variovorax*, *Afipia*, *Burkholderia*, and *Methylobacterium*. This group includes soil, water, symbiotic, and potentially pathogenic bacteria. *Rhizobium* and *Variovorax* are known for their nitrogen-fixing abilities, suggesting plant symbiosis. The presence of this diverse group indicates that *R. dominica* engages in a wide range of ecological interactions, including symbiosis with plants, interactions with other microorganisms, and various ecological niches [16,38,39].

The causal conclusions drawn from this analysis are as follows. First, the composition of the *R. dominica* microbiome is strongly linked to its living environment. Second, the presence of specific bacteria suggests that the microbiome of *R. dominica* may play roles in digesting organic matter, detoxification, and potentially protecting against pathogens. Third, the broad spectrum of bacteria in Group 4 suggests that *R. dominica* engages in numerous ecological interactions, including symbiosis with plants and other microorganisms. There are few reports on the role of the symbiotic microbiome of *R. dominica*, but limited information and data on other representatives of Coleoptera confirm the above observations [7,21,40].

Bacteria such as *Rhizobium* and *Variovorax* may indicate symbiotic relationships with plants, which could provide additional nitrogen sources to the insect. Pathogenic bacteria like *Pseudomonas* and *Serratia* may play roles in protecting against pathogens or pose a threat themselves, affecting the health and development of the insect. Regarding the environmental adaptation, the bacterial groups adapted to different ecological niches suggest that *R. dominica* is an insect with high ecological plasticity, allowing it to inhabit various environments.

The conducted research showed that changes in the microbiome, metabolism, and behaviour of insects are closely dependent on the specific chemical properties of grain cultivars, rather than the grain species as a whole (Figure 3 and Figure 5). Different cultivars within the same species uniquely affect the composition of the microbiome and metabolism of insects, which can lead to variations in feeding behaviours. For example, the analysis showed a different dominance of bacterial genera in some wheat and barley cultivars (Table 1). These changes influenced the metabolism of insects, suggesting that interactions between the diet and microbiome have direct impacts on their development and adaptation. Furthermore, different grain cultivars can affect insect behaviour, for instance, by reducing feeding efficiency through microbiome reorganization, which disrupts insect functioning and decreases their survival abilities.

The chemical properties of the different cereal cultivars were found to significantly affect the microbiome composition and feeding behaviour of *R. dominica*. Cereal cultivars with higher protein and water-soluble sugar (WSC) contents had a more diverse and active microbiome, which may indicate a better adaptation of the insects to such feeding conditions (Figure 5 and Table 1). *Pantoea* and *Ralstonia* bacteria were dominant in the first generation, while a greater microbial diversity was observed in the second generation, with the prominent presence of *Stenotrophomonas* and *Acinetobacter* (Table 1). Similar results for the effect of the grain composition on the microbiome of grain-feeding insects have been reported in the literature, indicating significant changes in the microbiome structure depending on the insect diet. Insects feeding on several food types and plant species will harbour more complex microbial communities [41]. In our study, such a trend can also be observed, as the microbiome of insects fed a different food than the basal food (from mass rearing) influenced the biodiversity of the microbiome in the second generation of insect life (Table 1). Pérez-Cobas et al. [42], Montagna et al. [20], Chouaia et al. [43], and Muturi et al. [44] confirmed that the diet can directly affect the composition of the microbiome, as bacteria capable of colonising in the insect gut can be inoculated from food, thereby changing the composition of the microbiome or affecting the promotion of the growth of specific bacteria. The Radek cultivar did not favour the development of a flexible microbiome, resulting in a rearrangement of the bacteriobiome towards a K-type developmental strategy, where bacterial taxa that reproduce strongly and use different autotrophic pathways thrive. Nevertheless, the bacteriobiome in this variant was characterised by a high aerobic chemoheterotrophic potential but a low overall chemoheterotrophic potential. As indicated in publications [40,45], the special symbiont species responsible for urea recycling and nitrogen recovery for amino acid construction is a crucial metabolic component in insects fed a diet low in nitrogen sources. The mechanism of urea utilisation is also linked to detoxification processes, which are important for insecticide degradation. Our results suggest that the complex chemical agents presented in the different cereal cultivars are pre-detoxified by the bacteriobiomes.

A complex analysis of the MetaCyc (MACADAM) database results [46] showed that almost all important OTUs specific to *R. dominica* are extensively involved in urea degradation and amino acid production. The urea degradation I pathway, in which urea-1-carboxylate and carbamate are intermediate metabolites with the participation of urea carboxylase and allophanate hydrolase [47], involves *Burkholderia*, *Pseudarthrobacter*, *Kocuria*, *Methylobacterium*, *Variovorax*, *Microvirga*, *Pseudomonas*, *Bosea*, *Escherichia*, *Corynebacterium*, *Xanthobacteraceae*, *Sphingobium*, *Massilia*, *Caulobacter*, *Xanthomonas*, *Alcaligenes*, *Acinetobacter*, *Pantoea*, *Paenibacillus*, *Cupriavidus*, *Ralstonia*, *Comamonadaceae*, *Serratia*, *Bacillus*, *Sphingomonas*, *Rhizobium*, and *Staphylococcus*. However, the process of direct degradation of urea to ammonium (urea degradation II) with the participation of urease [48] involves a complex of OTUs such as *Stenotrophomonas*, *Burkholderia*, *Sphingomonas*, *Pseudarthrobacter*, *Kocuria*, *Methylobacterium*, *Variovorax*, *Microvirga*, *Pseudomonas*, *Brevibacterium*, *Sporosarcina*, *Aureimonas*, *Bosea*, *Escherichia*, *Corynebacterium*, *Xanthobacteraceae*, *Sphingobium*, *Massilia*, *Xanthomonas*, *Acinetobacter*, *Pantoea*, *Paenibacillus*, *Moraxella*, *Cupriavidus*, *Ralstonia*, *Comamonadaceae*, *Ancylobacter*, *Serratia*, *Bacillus*, *Alcaligenes*, *Rhizobium*, and *Staphylococcus*. Kiefier et al. [45] also claimed that although most species closely related to *R. dominica* have unique obligate symbionts, these are often lacking in this species. The above scientists show that they are responsible for urea utilisation and amino acid production, especially lysine. Above, we indicated the possibility of urea utilisation by several facultatively symbiotic OTUs, and the accompanying additional study based on MACADAM data (MetaCyc) shows that bacteriobiomes have four pathways for L-lysine production, three of which are complete (Appendix A).

As studies on Coleoptera representatives show, numerous representatives of the genera *Pseudomonas* and *Sphingomonas* are responsible for the detoxification of undesirable substances in beetles [49], as an investigation of the MACADAM database (MetaCyc) also proved. In the present study, these two taxa were always dominant within the bacteriobiomes, despite apparent variations. A previous study showed that the wheat cultivar Rusałka and the barley cultivar Radek were the most resistant to *R. dominica* feeding [35]. In contrast, in the present study, the second-generation analysis showed a break in resistance, which was most noticeable in barley. The lack of significant differences in foraging on Radek compared to the other variants is closely related to the reorganisation of the bacteriome and the co-dominance of *Xanthomonas* sp., *Sulcia* sp. and *Alcaligenes* sp., which were much less important for the other bacteriobiomes. Of the three OTUs mentioned above, the genus *Sulcia* has the most important role in the adaptation to harsh feeding conditions. In many species, it is an obligate symbiont; in many pests, it is carried in the bacteriomes and develops during specific insect developmental stages. In this case, its development was probably triggered by poor living conditions in the first generation, and its development in the second generation broke the protective barrier of the Radek cultivar.

Bacteria of the genus *Sulcia* are responsible for synthesising at least eight amino acids necessary for host development [50]. It is puzzling that, together with *Sulcia*, the abundance of two other OTUs increased markedly. This may be due to the genome reduction of this symbiont and the inability to carry out several key metabolic pathways, including tryptophan production. Such was the case with Sulcia-CARI, where another symbiont, *Zinderia* sp., evolved together with *Sulcia* sp. to compensate for the metabolic deficiencies of the co-symbiont. Furthermore, both *Alcaligenes* and *Xanthomonas* can produce phytohormones, noting that *Xanthomonas* production of gibberellins may affect the reduction of jasmonic acid, thus lowering plant defences [51,52,53]. Given the above, it is quite likely that, in contrast to the permanent dominant species, that is, *Pantoea* sp., the occurrence of these three OTUs with a clear dominance of *Sulcia* sp. is a bioindicator indicating a permanent machinery of breaking varietal resistance in cereals by *R. dominica*.

After analysing the results obtained, it should also be noted that in the case of *R. dominica*, its microbiome can effectively support the digestion of the insect, but is not obligatory because by eliminating all its microbes, the insect is able to continue to survive [7,27]. So, the microbiome performs only functions that support and facilitate the functioning and life of this storage pest, underscoring the practical importance of understanding the insect’s microbiome.

## 4. Materials and Methods

### 4.1. Experimental Design

The experiment involved the following steps: (1) selection of cereal cultivars based on the developmental parameters of the *R. dominica*; (2) establishment and standardisation of mass breeding of *R. dominica*; (3) chemical analyses of the grains; (4) experimentation with different feed variants; (5) DNA isolation and molecular analyses; (6) analysis of the biochemical properties of the insects; (7) assessment of grain degradation; and (8) statistical analysis of the results using XLSTAT software. The detailed methodology is shown in Figure 7.

### 4.2. Experiment with Different Feed Variants

The lesser grain borer (*R. dominica*) used in the experiment was obtained from mass breeding carried out at the Department of Entomology, Phytopathology, and Molecular Diagnostics at UWM in Olsztyn. Twenty 1–2-day-old adult beetles of *R. dominica* with a 1:1 sex ratio [23] were placed on 500 g of grain in 1-liter glass containers. A chiffon mesh protected the openings of the boxes, allowing air to enter and providing ventilation. The experiment involved using different cultivars of wheat and barley. Specifically, three wheat cultivars (Rusałka, Varius, and Impresja) were chosen, all belonging to the high-quality bread cultivars in the A technological group. Additionally, three cultivars of barley (Radek, Trofeum, and Ismena) were included, and they are classified as fodder type (P) [54]. The cultivars were chosen based on preliminary research involving ten wheat and ten barley cultivars. These cultivars are the focus of a separate study. Through our analysis of the survival data (number of progeny) of the lesser grain borer on the tested cultivars, we have categorised the Radek barley cultivar as resistant, the Ismena cultivar as having intermediate resistance, and the Trofeum cultivar as having low resistance. As for the wheat cultivars, the Rusałka cultivar demonstrated the highest level of resistance, the Varius cultivar showed moderate resistance, and the Impresja cultivar was classified as having low resistance [35]. Insects without food were the background. Each treatment was carried out in three replicates. Cultivation took place under controlled conditions of temperature (30 °C) and humidity (70%) in a breeding chamber (Sanyo MLR—350 H—Sanyo Electric Co., Ltd., Osaka, Japan). The incubation parameters were associated with the best habitat for developing *R. dominica* [55]. Adult insects were selected for analysis after 20 days (first generation; I) and 40 days (second generation; II). The digestive systems of 10 specimens were prepared in a laminar chamber using binoculars and sterile instruments. The insects were anesthetised, sterilised, and homogenised as per the methodology in Kosewska et al. [7].

### 4.3. Chemical Analyses of Grains

The dry mass was determined using the drying method. The raw ash content was determined by burning an air-dried sample. The total protein content was determined based on the total nitrogen (N) content determined by the Kjeldahl method following the Polish Standard PN-ISO 5983.2000. Crude fat was extracted using a Soxhlet extractor, following standard PN-ISO 6492.2005. Fatty acid methyl esters were prepared using a modified Peisker method to determine the fatty acid composition. The anthrone method was used to determine the water-soluble sugar (WSC) content [56]. The crude fibre content was determined by the classic Henneberg–Stohmann method for determining the raw fibre content of feed following the PN–EN ISO 6865.2002 standard. The antioxidant capacity of samples was determined according to the Benzie and Strain method [57] with some modifications [25]. All methods used to determine individual chemical components have been thoroughly described in the publication by Kosewska et al. [25].

### 4.4. Biochemical Properties of Insects

The digestive tracts of mature insects were assessed for their biochemical activity and carbon source utilisation using API ZYM and API 20 NE tests (Biomerieux, Marcy-l'Étoile, France). Each variant was represented by approximately 100 mg samples, which were then homogenised in vials with glass beads and 1 mL of peptone water. Homogenisation was carried out using a TissueLyser LT bead homogeniser (Qiagen, Hilden, Germany) with 30 oscillations for 5 min. After homogenisation, the resulting suspension was diluted at a 1:10 ratio in peptone water enriched with 1% TSB (Tryptic soy broth, Merck, Germany) and then incubated at 30 °C for 4 h before being used in the API tests, all in line with the manufacturer’s guidelines. The samples for the biochemical and carbon source tests were also incubated at 30 °C for 24 h [58]. The genetic material was isolated using the Soil Purification Kit (EURx), and sequencing was conducted with the “16S Barcoding Kit 1–24 (SQK-16S024)”. Bacterial 16S rRNA amplification employed 24 barcoding primers and the HiFi ToughMix repliQa polymerase (Quantabio, Beverly, MA, USA). The library construction and pooling procedures adhered to the guidelines provided by the polymerase supplier (“Rapid 16S Metagenomic Library Preparation for Oxford Nanopore Technologies (ONT)^®^ Platform”; www.quantabio.com/product/repliqa-hifi-toughmix/ (accessed on 5 April 2024)). Subsequent sequencing took place in a flongle flow cell utilising the MinKnown software (https://nanoporetech.com/news/news-introducing-new-minknow-app (accessed on 31 July 2024)). Our DNA data were submitted to the Sequence Read Archive (SRA) of NCBI, BioProject ID: PRJNA1161390. The obtained data, comprising approximately 62,000 reads, underwent analysis via the EPI2ME platform (Metrichor™ Ltd., Oxford, UK) utilising WIMP (What’s In My Pot) workflows. The postprocessing and OTU table generation were carried out using the BugSeq bioinformatics platform (https://bugseq.com/ (accessed on 15 April 2024)).

### 4.5. Statistical Calculations

The chemical properties of the grain and developmental characteristics were tested for normal distributions and homogeneity of variance. Differences between the results were determined using ANOVA (Tukey’s test) or the Kruskal–Wallis test (Dunn’s test). Additionally, NMDS was performed within species based on the Bray–Curtis dissimilarity matrix. The taxonomic diversities of the analysed OTUs were determined with the use of Simpson dominance (λ), Shannon diversity (H’), and Pielou’s evenness (J’) indices. Diversity indices and rarefaction curves were obtained by the PAST 4.13 program [59]. Dominant classes were determined according to previous work [60]. In order to determine the variation between bacterial biomes and the effects of factors on diversity, PERMANOVA and ANOSIM tests (XLSTAT—R vegan) were applied. The MACADAM database [46] was used to explore predicted microbial community functions and FAPROTAX was used to assess metabolic function [61]. Agglomerative Hierarchical Clustering (AHC) dissimilarities of microbiomes were calculated based on the Bray–Curtis method, and dendrograms were constructed based on Ward’s method. Principal component analysis (PCA) was performed based on the Pearson similarity matrix, including microbiome data (OTUs > 1%), metabolic activity, microbial predicted physiological properties, and the chemical parameters of grain. Heatmaps were constructed for predicted metabolic functions, and metabolic activity properties were constructed based on N-standardized data. Partial Least Squares Path Modelling (PLS-PM) was performed using standardised weight data for manifest variables, with the centroid as the internal estimate and 95% confidence intervals (bootstrap: 100 resampling). Statistical calculations were made with the XLSTAT program [62]. The network analysis was performed using the AxisForce 2 algorithm with LinLog mode [63] based on Spearman’s correlation matrix for edges and the cumulative percentage of each variable for nodes [64].

## 5. Conclusions

The study found that the chemical properties of different cereal species, such as wheat and barley and their cultivars, significantly impact the development and metabolism of the bacteriobiome of *R. dominica* and its foraging. The results showed that grains with higher levels of total protein (TP) and water-soluble sugars (WSCs) were more attractive to the insect, resulting in more intensive feeding and development of these pests. Wheat cultivars such as Varius, which are characterised by a high protein content, were preferred by *R. dominica*, confirming the hypothesis that the grain chemical composition influences insect foraging intensity. The selection of cultivars of different cereal species with lower TP and WSC contents could be an effective strategy for reducing the population of this pest. An analysis of bacteriobiome diversity indices revealed that the dominance of bacteria from the genera *Pantoea* and *Ralstonia* was a characteristic of most cultivars in the first generation. However, we observed a greater microbial diversity in the second generation, with the prominent presence of *Stenotrophomonas* and *Acinetobacter* bacteria. Notably, grains of the Radek cultivar showed a low potential for utilisation of different carbon sources, suggesting that a low-nitrogen diet may increase the involvement of bacterial symbionts in urea recycling processes and the production of amino acids necessary for insect growth. The composition of the *R. dominica* microbiome is related to the insect diet. Bacteria from the genera *Pantoea*, *Ralstonia*, *Stenotrophomonas*, and *Acinetobacter* play crucial roles in insect metabolism, influencing their ability to adapt to different dietary conditions. The linoleic acid (C18:2) content of grains may influence the rearrangement of the insect microbiome, which in turn may reduce their foraging efficiency. In addition, the presence of antioxidants in the grains showed a trend of effects on the microbiome and growth of *R. dominica*. Rearrangements of the microbiome structure contributed to an increase in the share of entomopathogens, which led to the displacement of lysine-producing symbionts. To answer how this happens and what the sequence of events is in this complex system, comprehensive metatranscriptomic and metabolomic analyses of both the insect and microorganisms are necessary. The effect of bacteria-derived lysine on insect metabolism suggests that perturbations in lysine production may negatively affect *R. dominica* foraging. Chemical properties of the grains, such as higher linoleic acid and antioxidant contents, can therefore be used to control the population of this pest by influencing its microbiome and foraging.

The lack of significant correlations at the species level, as shown by the ANOSIM results, is crucial because it indicates substantial differences between cultivars within each species, which complicates the identification of general patterns between species. These results emphasize that research should focus on specific cultivars and their unique traits rather than general differences between grain species. This approach will provide a better understanding of how specific chemical properties of cultivars affect the microbiome, metabolism, and functioning of insects. The results for cultivars were significant, highlighting notable differences in the microbiome structure and insect metabolism, which were also confirmed by structural equation plots. This finding has important implications for future research, suggesting that identifying specific cultivar traits may be more valuable than species-level studies.

This study’s results suggest that by manipulating the chemical properties of the grain and understanding the interactions between the diet, microbiome, and insect physiology, we can potentially develop new, more sustainable methods of controlling *R. dominica*.

Growing cultivars of selected cereal species with specific chemical properties can significantly change agricultural practices. First and foremost, selecting and cultivating cereal cultivars that are naturally less attractive to storage pests, such as *R. dominica*, can reduce the need for synthetic pesticides, reducing storage losses and contributing to more sustainable agricultural production. This can also have direct environmental impacts, reducing soil and water pollution levels, as well as reducing negative impacts on beneficial organisms. It should also be emphasised that the use of resistant cereal cultivars will support the development of organic farming. In addition, it can increase farm efficiency by reducing the costs associated with plant protection and increasing the stability of the yields obtained.

It should also be stressed that reducing losses during grain storage is extremely important in the fight against hunger, especially in the poorest regions of the world, where inefficient storage and a lack of resources for pest control lead to more significant losses. The introduction of naturally pest-resistant cultivars could significantly increase food availability, which is particularly important where food is scarce. This will contribute to the creation of stable food stocks, which are invaluable during crises, and will strengthen social stability.

## Figures and Tables

**Figure 1 ijms-25-10130-f001:**
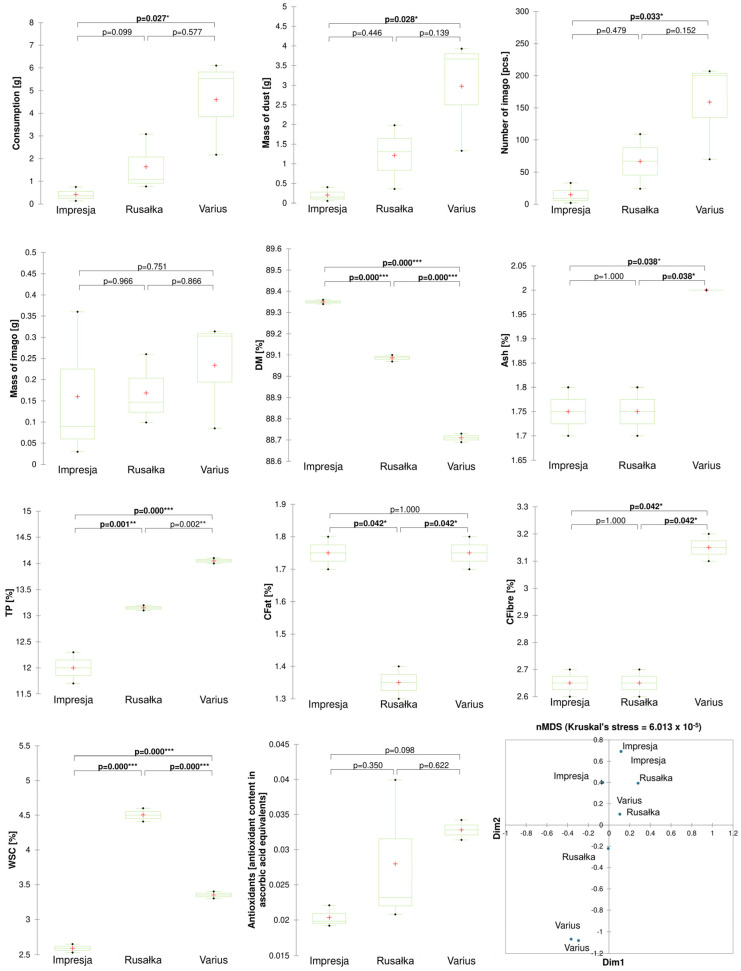
Boxplots of *R. dominica* developmental and wheat chemical parameters, depending on the cultivar used, and Bray–Curtis hierarchical clustering and NMDS show dissimilarity between treatments. Significance levels are denoted by asterisks as follows: * *p* < 0.05, ** *p* < 0.01 and *** *p* < 0.001. Abbreviations: DM—dry matter, Ash—crude ash, TP—total protein, CFat—crude fat, CFibre—crude fibre, WSC—water-soluble carbohydrate.

**Figure 2 ijms-25-10130-f002:**
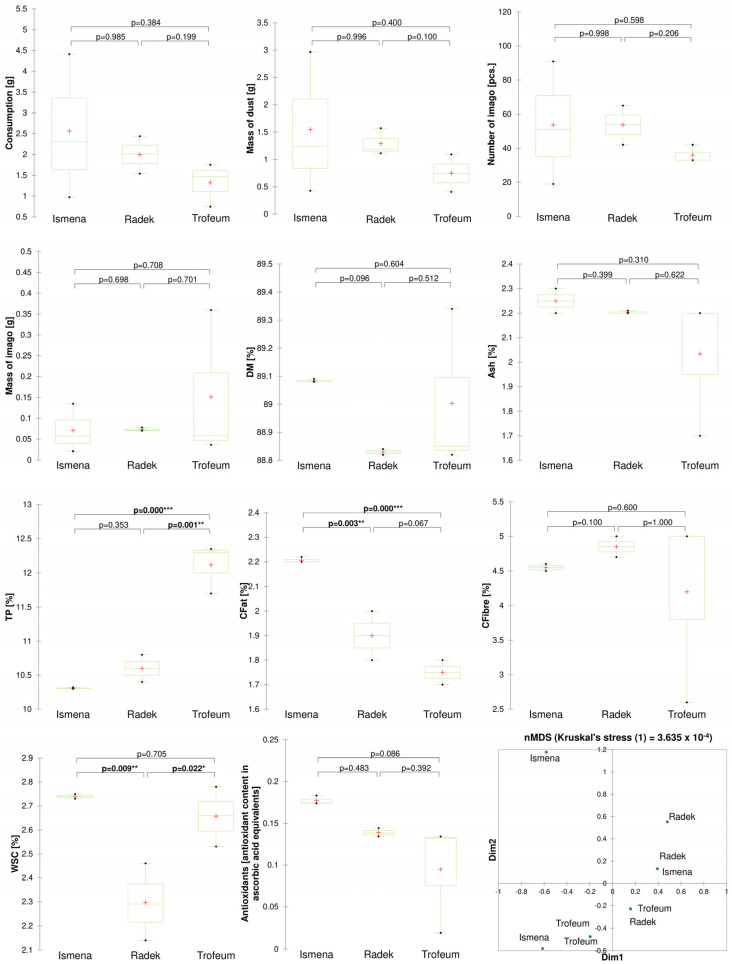
Boxplots of *R. dominica* developmental parameters and barley chemical parameters, depending on the cultivar used, and Bray–Curtis hierarchical clustering and NMDS show dissimilarity between treatments. Significance levels are denoted by asterisks as follows: * *p* < 0.05, ** *p* < 0.01 and *** *p* < 0.001. Abbreviations: DM—dry matter, Ash—crude ash, TP—total protein, CFat—crude fat, CFibre—crude fibre, WSC—water-soluble carbohydrate.

**Figure 3 ijms-25-10130-f003:**
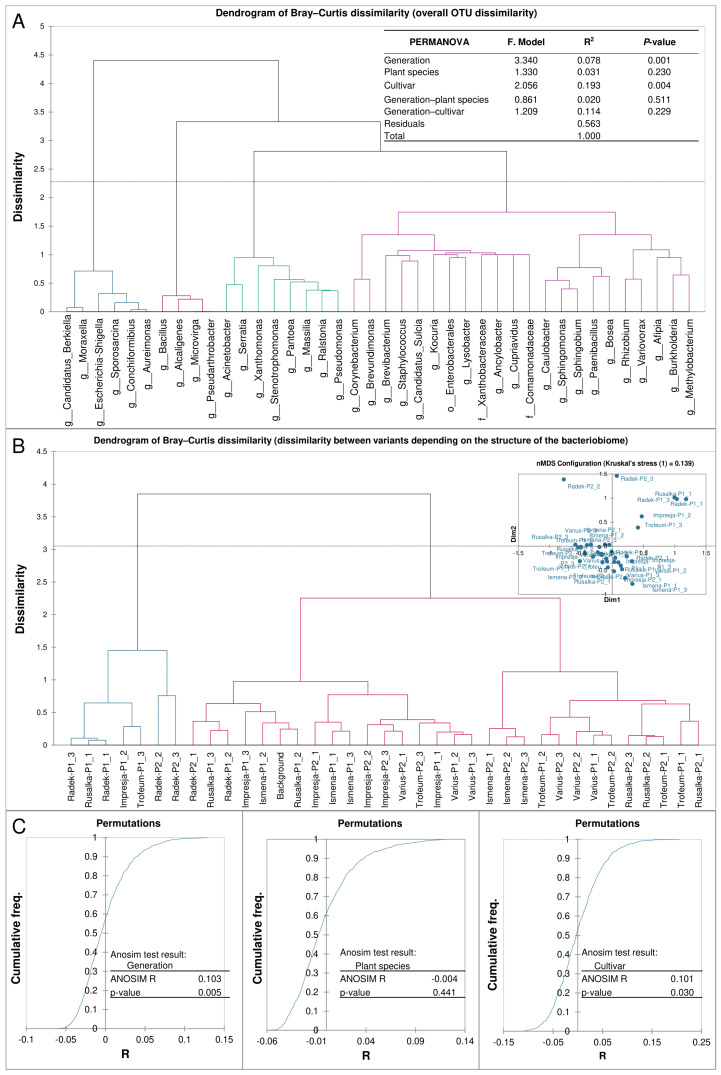
Variation of the bacteriome structure depending on the factors applied. (**A**) Grouping of bacterial genera based on the overall dissimilarity of OTUs. (**B**) Dissimilarity between the different variants of cultivars and generations applied depending on the structure of the *R. dominica* bacteriome, including an NMDS plot visualising the dissimilarity between variants. (**C**) ANOSIM test results showing the effects of the generation, plant species, and cultivar on the bacterial community structure of *R. dominica*.

**Figure 4 ijms-25-10130-f004:**
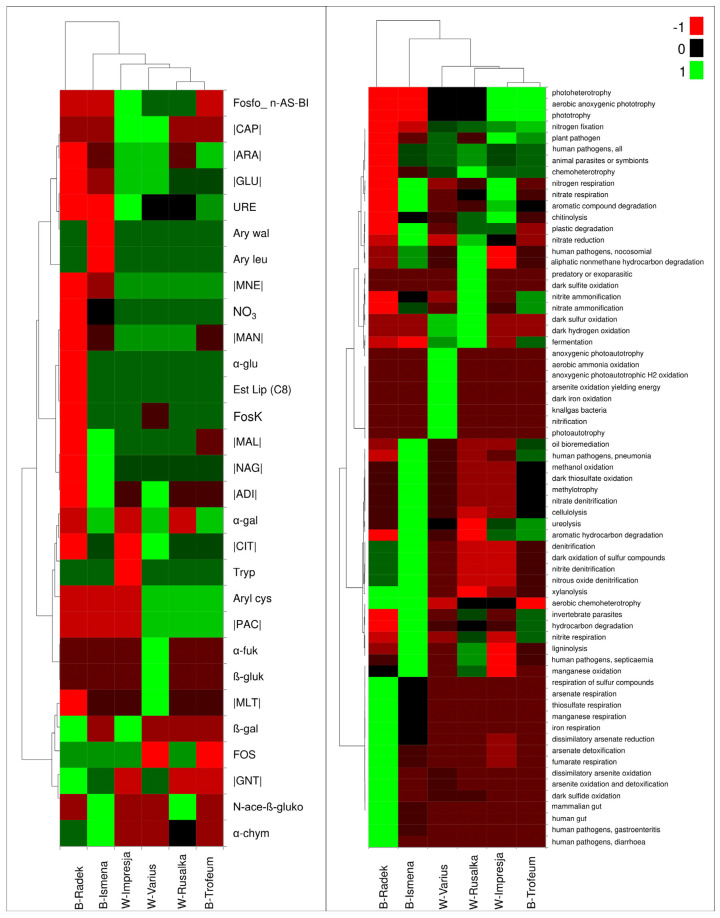
Heatmap illustrating the differentiation of bacteriobiome metabolic traits (**left** side) and potential bacteriome traits (multitraits; **right** side), depending on the cultivar (abbreviations: B—barley, W—wheat, Fosfo_n-AS-BI—naphthyl-AS-BI phosphohydrolase, |CAP|—decanoic acid utilisation, |ARA|—L-arabinose utilisation, |GLU|—D-glucose utilisation, URE—urease activity, Ary wal—valine arylamidase, Ary leu—leucine arylamidase, |MNE|—D-mannose utilisation, NO3—nitrate reduction, |MAN|—D-mannitol utilisation, α-glu—α-glucosidase, Est Lip (C8)—esterase lipase (C8), FosK—acid phosphatase, |MAL|—utilisation of D-maltose, |NAG|—N-acetyl-glucosamine utilisation, |ADI|—utilisation of adipic acid, α-gal—α-galactosidase, |CIT|—trisodium citrate utilisation, Tryp—trypsin, Aryl cys—cystine arylamidase, |PAC|—utilisation of phenylacetic acid, α-fuk—α-fucosidase, β-gluk—β-glucosidase, |MLT|—malate utilisation, β-gal—β-galactosidase, FOS—alkaline phosphatase, |GNT|—potassium gluconate utilisation, N-ace- β-gluko—N-acetyl-β-glucosaminidase, α-chym—α-chymotrypsin).

**Figure 5 ijms-25-10130-f005:**
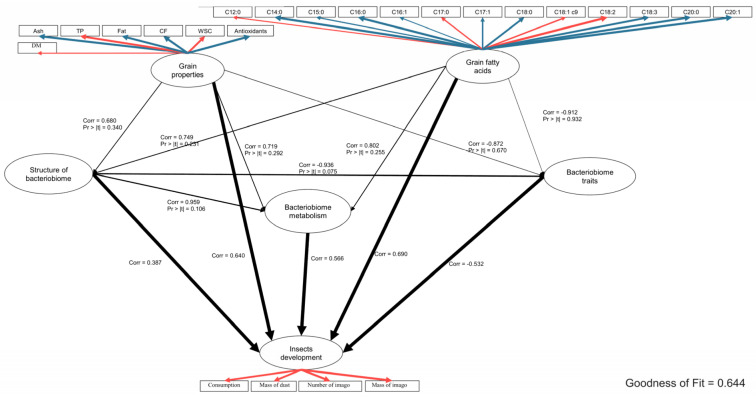
Partial Least Squares Path Modelling (PLS-PM) showing the mutual influences of groups of variables (all results included). For grain properties, grain fatty acid contents and insect development, internal correlations between individual variables manifested in a given group are indicated. The thickness of the arrow indicates the p-value: thick, *p* < 0.05; mean, *p* = 0.1 (trend); thin, *p* < 0.1. Red arrow colour—positive correlation; blue arrow colour—negative correlation. Abbreviations: Ash—crude ash, TP—total protein, Fat—crude fat, CF—crude fibre, WSC—water-soluble carbohydrate, DM—dry matter.

**Figure 6 ijms-25-10130-f006:**
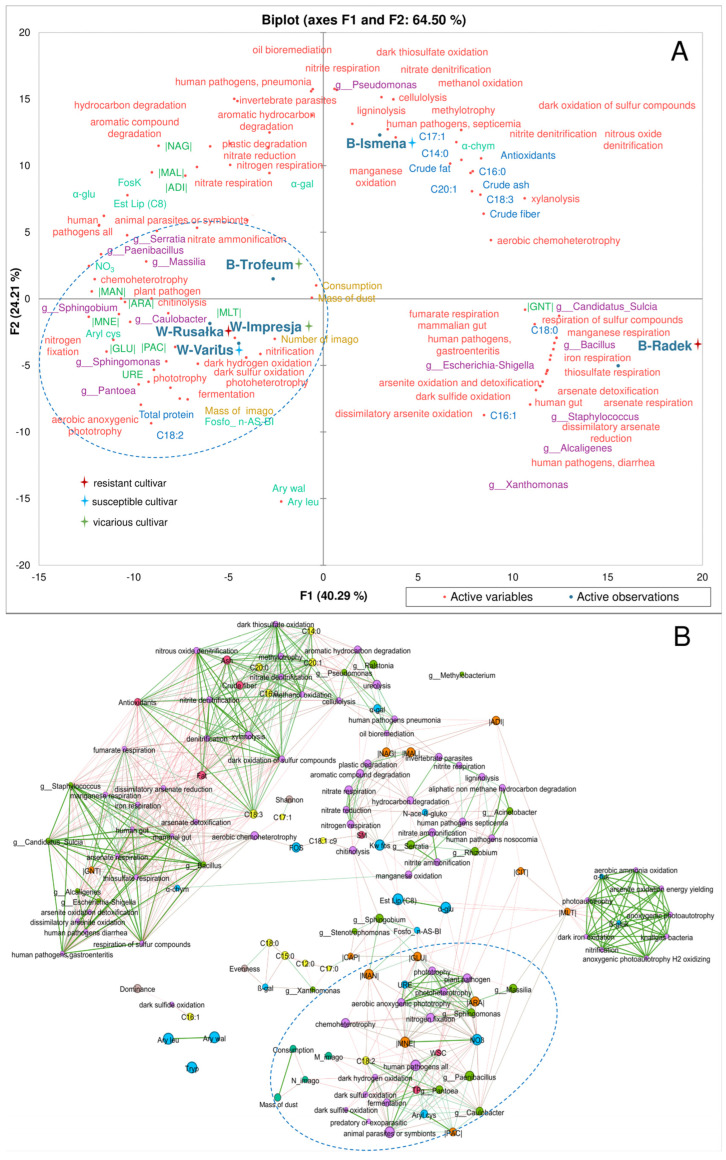
PCA (principal component analysis) (**A**) showing relationships between substitutes and variants, and a co-occurrence network analysis (**B**) showing correlations between all characteristics included in the study (r < 0.75). Abbreviations: NO3—nitrate reduction, URE—urease activity, |GLU|—D-glucose utilisation, |ARA|—L-arabinose utilisation, |MNE|—utilisation of D-mannose, |MAN|—utilisation of D-mannitol, |NAG|—N-acetyl-glucosamine utilisation, |MAL|—utilisation of D-maltose, |GNT|—potassium gluconate utilisation, |ADI|—utilisation of adipic acid, |MLT|—malate utilisation, |PAC|—utilisation of phenylacetic acid, Est Lip (C8)—esterase lipase (C 8), Ary leu—leucine arylamidase, Ary wal—valine arylamidase, Aryl cys—cystine arylamidase, α-chym—α-chymotrypsin, FosK—acid phosphatase, Fosfo_ n-AS-BI—naphthyl-AS-BI phosphohydrolase, α-gal—α-galactosidase, α-glu—α-glucosidase.

**Figure 7 ijms-25-10130-f007:**
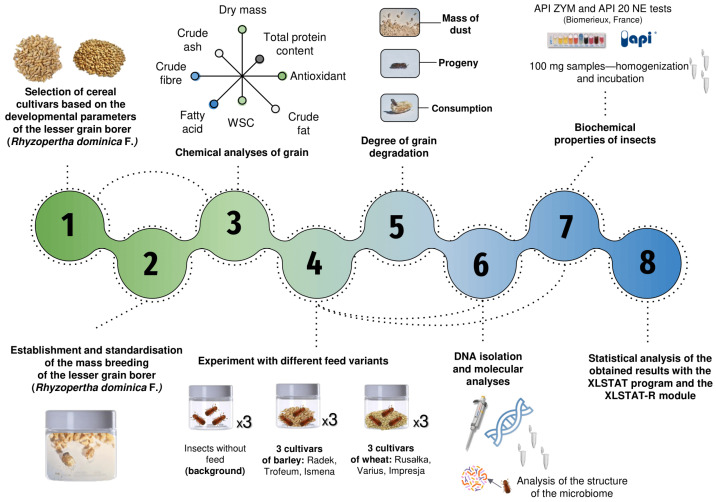
Schematic representation of the experimental workflow for assessing the impacts of different cereal cultivars on the lesser grain borer (*Rhyzopertha dominica* F.).

**Table 1 ijms-25-10130-t001:** Structure of the *R. dominica* bacteriome and indicators of microflora diversity identified (percentage of bacterial genera identified in the digestive system of *R. dominica* feeding on different wheat and barley cultivars). Values sharing the same letters (a, ab, b) are not significantly different from each other according to Tukey's test.

Generation	Generation I							Generation II					
Species of the	Background without Feed	Wheat			Barley			Wheat			Barley		
Cultivar		Impresja	Rusałka	Varius	Trofeum	Radek	Ismena	Impresja	Rusałka	Varius	Trofeum	Radek	Ismena
g__Pantoea	23.3	18.9	18.6	24.2	29.2	12.9	12.3	19.7	35.3	36.0	28.0	10.8	10.1
g__Ralstonia	21.6	14.3	11.4	27.7	8.9	6.9	22.8	19.9	3.3	11.0	19.2	5.1	15.6
g__Staphylococcus	6.0	20.3	28.7	0.0	14.5	54.9	0.7	0.0	0.0	0.0	0.0	7.8	0.4
g__Pseudomonas	9.3	17.1	9.3	11.0	10.1	6.4	11.2	6.3	5.4	8.8	10.8	5.5	22.6
g__Stenotrophomonas	10.8	6.4	7.9	11.4	0.9	5.7	25.3	22.3	4.7	6.1	1.2	1.8	5.4
g__Massilia	7.6	5.3	4.8	10.3	8.5	2.2	7.4	9.5	5.2	12.4	9.9	1.6	8.1
g__Acinetobacter	0.0	0.3	0.5	0.6	12.8	0.8	1.4	1.4	21.1	7.4	7.1	4.0	13.8
g__Serratia	0.0	0.4	0.7	1.7	6.4	0.6	0.0	10.3	17.7	9.1	14.1	0.7	11.1
g__Xanthomonas	0.0	3.2	8.8	6.5	0.7	4.9	7.2	2.3	0.0	2.3	1.8	14.3	0.0
g__Sphingomonas	6.6	4.1	2.5	3.0	1.2	1.1	2.2	3.2	2.2	2.0	3.1	1.2	1.2
g__Candidatus_Sulcia	0.0	0.0	0.0	0.0	0.5	0.0	0.6	0.0	0.0	0.0	0.0	19.6	6.4
g__Sphingobium	4.4	2.5	0.6	0.6	0.4	1.3	1.0	1.4	1.1	1.2	0.7	0.4	0.9
o__Enterobacterales	7.9	2.2	4.5	0.3	0.0	0.6	3.6	0.0	0.0	0.0	0.0	0.0	0.0
g__Caulobacter	0.0	1.8	0.3	1.4	0.5	0.7	0.7	0.5	0.7	1.3	0.9	0.0	0.4
g__Paenibacillus	0.0	0.5	0.3	0.8	0.7	0.0	1.1	0.8	1.0	1.0	1.1	0.0	0.7
g__Brevibacterium	0.0	0.0	0.0	0.0	3.1	0.0	0.0	0.0	0.0	0.0	0.0	0.0	0.0
g__Bosea	0.0	0.7	0.0	0.0	0.2	0.0	0.2	0.3	0.3	0.4	0.3	0.4	0.6
g__Rhizobium	0.0	0.3	0.0	0.0	0.0	0.7	0.6	0.0	0.5	0.3	0.5	0.0	0.3
g__Variovorax	0.0	0.0	0.0	0.0	0.2	0.0	0.0	0.0	0.7	0.2	0.8	0.0	0.9
g__Burkholderia	2.7	0.0	0.0	0.0	0.3	0.0	0.0	1.0	0.2	0.5	0.0	0.0	0.4
g__Bacillus	0.0	0.8	0.0	0.0	0.0	0.0	0.0	0.0	0.0	0.0	0.0	4.2	0.9
g__Cupriavidus	0.0	0.0	0.0	0.0	0.0	0.0	1.4	0.0	0.0	0.0	0.0	0.0	0.0
g__Methylobacterium	0.0	0.4	0.0	0.3	0.2	0.0	0.0	1.2	0.0	0.0	0.0	0.0	0.0
g__Afipia	0.0	0.6	0.2	0.0	0.0	0.0	0.0	0.0	0.0	0.0	0.0	0.0	0.0
g__Escherichia-Shigella	0.0	0.0	0.0	0.0	0.0	0.0	0.0	0.0	0.0	0.0	0.0	2.4	0.0
g__Corynebacterium	0.0	0.0	0.4	0.2	0.0	0.0	0.0	0.0	0.0	0.0	0.0	0.0	0.2
g__Brevundimonas	0.0	0.0	0.4	0.0	0.0	0.0	0.0	0.0	0.0	0.0	0.5	0.0	0.0
g__Alcaligenes	0.0	0.0	0.0	0.0	0.0	0.0	0.0	0.0	0.0	0.0	0.0	5.1	0.0
g__Sporosarcina	0.0	0.0	0.0	0.0	0.0	0.0	0.0	0.0	0.0	0.0	0.0	1.7	0.0
f__Comamonadaceae	0.0	0.0	0.0	0.0	0.2	0.0	0.0	0.0	0.0	0.3	0.0	0.0	0.0
g__Conchiformibius	0.0	0.0	0.0	0.0	0.0	0.0	0.0	0.0	0.0	0.0	0.0	1.4	0.0
g__Ancylobacter	0.0	0.0	0.0	0.0	0.4	0.0	0.0	0.0	0.0	0.0	0.0	0.0	0.0
g__Aureimonas	0.0	0.0	0.0	0.0	0.0	0.0	0.0	0.0	0.0	0.0	0.0	1.3	0.0
g__Lysobacter	0.0	0.0	0.0	0.0	0.0	0.0	0.4	0.0	0.0	0.0	0.0	0.0	0.0
f__Xanthobacteraceae	0.0	0.0	0.0	0.0	0.0	0.0	0.0	0.0	0.5	0.0	0.0	0.0	0.0
g__Microvirga	0.0	0.0	0.0	0.0	0.0	0.0	0.0	0.0	0.0	0.0	0.0	4.7	0.0
g__Pseudarthrobacter	0.0	0.0	0.0	0.0	0.0	0.0	0.0	0.0	0.0	0.0	0.0	4.7	0.0
g__Candidatus_Berkiella	0.0	0.0	0.0	0.0	0.0	0.0	0.0	0.0	0.0	0.0	0.0	0.8	0.0
g__Moraxella	0.0	0.0	0.0	0.0	0.0	0.0	0.0	0.0	0.0	0.0	0.0	0.7	0.0
g__Kocuria	0.0	0.0	0.0	0.0	0.0	0.2	0.0	0.0	0.0	0.0	0.0	0.0	0.0
	Eudominant			Dominant			Subdominant			Occasional		Casual	
	>10.01%			5.1–10.0%			2.1–5.0%			1.1–2.0%		<1.0%	
Dominance_D		0.1832 ab	0.2185 ab	0.2104 ab	0.2398 a	0.6804 a	0.2033 ab	0.1595 ab	0.2579 a	0.1986 ab	0.2256 ab	0.2341 ab	0.1322 b
Shannon_H		2.0590 ab	1.7480 b	1.8110 b	1.8460 b	0.7871 b	1.8510 b	2.0770 ab	1.7390 b	2.0090 ab	1.7700 ab	1.9490 ab	2.2490 a
Evenness_e^H/S		0.5848 ab	0.5223 ab	0.5982 ab	0.5278 b	0.2632 b	0.5292 ab	0.6445 a	0.4831 b	0.4973 ab	0.5703 ab	0.5866 ab	0.4989 ab

## Data Availability

Our DNA data were submitted to the Sequence Read Archive (SRA) of NCBI, BioProject ID: PRJNA1161390.

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
