# Peer review of "Influence of the Chemical Properties of Cereal Grains on the Structure and Metabolism of the Bacteriome of Rhyzopertha dominica (F.) and Its Development: A Cause–Effect Analysis"

_ijms, 2024, doi:10.3390/ijms251810130_

Round 1

Reviewer 1 Report

Comments and Suggestions for Authors

The paper "Influence of Chemical Properties of Cereal Grains on the Structure and Metabolism of the Bacteriome Rhyzopertha dominica (F.) and Its Development: A Cause-Effect Analysis", propose an interesting study on the impact of the feed on the  structure and metabolism of the bacteriome of the R. dominca. The paper was well structured.The introduction, gives readers  an exhaustive overview of the topic, including a quantification of economic and social losses, to highlight the importance of the fight against pest.The Authors offer an comprehensive description the R. dominica introducing the topic of the paper clearly .The materials and methods were well described as results. Overall, the paper can be considered an excellent scientific examination of the impact of feed on the microbiome and on the development of R. dominica.  It is a valid support for future studies that address current global needs to reduce the use of chemicals and provide a more sustainable response to natural insect control.Despite, I would suggest to the Authors to emphasiser  in the conclusions how reducing the grain storage losses can be a strategic action in the fight against hunger even in the poorest countries, where storage-related losses are even more severe, than the relevance of this research field. 

Author Response

Comments 1:

The paper "Influence of Chemical Properties of Cereal Grains on the Structure and Metabolism of the Bacteriome Rhyzopertha dominica (F.) and Its Development: A Cause-Effect Analysis", propose an interesting study on the impact of the feed on the  structure and metabolism of the bacteriome of the R. dominca. The paper was well structured.The introduction, gives readers  an exhaustive overview of the topic, including a quantification of economic and social losses, to highlight the importance of the fight against pest. The Authors offer an comprehensive description the R. dominica introducing the topic of the paper clearly.The materials and methods were well described as results. Overall, the paper can be considered an excellent scientific examination of the impact of feed on the microbiome and on the development of R. dominica.  It is a valid support for future studies that address current global needs to reduce the use of chemicals and provide a more sustainable response to natural insect control. Despite, I would suggest to the Authors to emphasiser in the conclusions how reducing the grain storage losses can be a strategic action in the fight against hunger even in the poorest countries, where storage-related losses are even more severe, than the relevance of this research field. 

Response 1

Thank you very much for your thoughtful and detailed review of our article. We greatly appreciate the positive feedback on the structure and content of our paper, as well as the recognition of its potential contribution to future research focused on sustainable pest control.

We would also like to thank you for your valuable suggestion to emphasize the strategic importance of reducing grain storage losses as a means to combat hunger, particularly in the poorest regions where such losses can have devastating effects. We agree that this aspect is crucial and could indeed strengthen the relevance of our research. In light of your suggestion, we will make sure to emphasize this point more clearly in the conclusion of our article.

Your input will help us improve our work and ensure that it not only contributes to the scientific community but also addresses broader global challenges. Thank you again for your constructive and insightful comments.

The following section has been added to the conclusion section:

"It should also be stressed that reducing losses during grain storage is extremely important in the fight against hunger, especially in the poorest regions of the world, where inefficient storage and lack of resources for pest control lead to more significant losses. The introduction of naturally pest-resistant cultivars could significantly increase food availability, which is particularly important where food is scarce. This will contribute to the creation of stable food stocks, which are invaluable during crises and will strengthen social stability."

Reviewer 2 Report

Comments and Suggestions for Authors

Areas for improvement:

Please split the overly complex objective into two tasks to make it clearer.

Explain what specific parameters for the development of R. dominica were considered in the selection of cereal varieties.

Describe in detail the growing conditions such as temperature, humidity, type of containers, etc.

Please add more detail on the scale of losses caused by R. dominica.

Please describe in more detail what specific changes were observed in the insect microbiome and metabolism in response to different grain varieties.

Although the ANOSIM results did not show significant correlations at the species level, it is worth explaining why these results are important and what implications they have for further research.

Please explain in more detail why the microbiome is so important, giving examples or references to the literature.

It is worth adding what specific changes have been observed in the microbiome, metabolism and behaviour of insects.

It is worth explaining what is meant by a lack of correlation at the species level and why this is important.

Please explain why the relationships between C18:2 fatty acid, entomopathogenic bacteria, impaired nitrogen cycle, lysine production of bacterial origin, and insect feeding are important and how they affect insect behaviour.

It is worth adding what specific changes in agricultural practices may result from these findings and what agricultural and environmental benefits may result from breeding resistant cereals.

Overall, the article makes a valuable contribution to the understanding of the interaction between grain chemical properties and the storage pest microbiome, which may lead to the development of more sustainable methods to control these storage pests.

Minor linguistic corrections

4.4 Biochemical properties of insects

peptone wateR -> peptone water

he genetic material -> The genetic material

OUT table -> OTU table

5. Conclusions

resulting in more intensive feeding and more intense development -> resulting in more intensive feeding and development

Wheat cultivars such as Varius, characterised by high protein content, were preferred by the R. dominica -> Wheat cultivars such as Varius, characterised by high protein content, were preferred by R. dominica

suggesting that a low-nitrogen diet may lead to greater involvement of bacterial symbionts in urea recycling processes and production of amino acids necessary for insect growth -> suggesting that a low-nitrogen diet may increase the involvement of bacterial symbionts in urea recycling processes and the production of amino acids necessary for insect growth

Author Response

Comments 1: Please split the overly complex objective into two tasks to make it clearer.

Response 1: Thank you for your comment. To make the aim of the study clearer, it has been rewritten (The verification of the adopted research hypothesis was carried out by:

- Confirming the influence of the chemical properties of the tested wheat and barley cultivars on the composition and activity of the symbiotic microbiome of R. dominica;

- Confirming the close correlation between the process of food digestion by R. dominica and the enzymatic activity and interaction with the symbiotic microbiome of this insect)

Comments 2: Explain what specific parameters for the development of R. dominica were considered in the selection of cereal varieties.

Response 2: Thank you for your comment. In selecting the cereal cultivars for our study, we were guided by several key parameters for the development of Rhyzopertha dominica. First of all, we took into account factors such as the number of the progeny generation (adult insects), which is directly related to foraging intensity, and the weight of the resulting dust. These parameters are key, as they directly reflect the insect's ability to survive, reproduce and further develop on a given cereal variety.

Added to text: number of progeny (Through our analysis of the survival data (number of progeny) of the lesser grain borer on the tested cultivars......)

Comments 3: Describe in detail the growing conditions such as temperature, humidity, type of containers, etc.

Response 3: Thank you for your comment. In our experiment, the growth conditions for Rhyzopertha dominica were carefully controlled to ensure an optimal environment for the development of this pest. Insects were reared at a temperature of 30°C and a relative humidity of 70%, which corresponds to conditions favourable for the growth of this species in grain stores. The research was conducted in a Sanyo MLR-350 climate chamber. 1 litre glass containers were used to store and rear the insects, which were sealed with a chiffon mesh to ensure adequate ventilation and air access, while preventing insect escape. Each container contained 500 grams of grain of a selected wheat or barley cultivar, on which 20 adults of R. dominica were placed (in an equal sex ratio of 1:1). The containers were kept under controlled incubation conditions for a suitable period of time to allow full development of the insect generation.

For a detailed description of these conditions, see ‘Experiment with different feed variants’.

Comments 4: Please add more detail on the scale of losses caused by R. dominica.

Response 4: Thank you for your comment. To better illustrate the magnitude of losses caused by Rhyzopertha dominica, we have added numerical values to the description of grain weight losses. For wheat, information has been added to the sentence: “Varius cultivar, where the highest grain weight loss was recorded (4.6 g), and the Impresja cultivar, which had the lowest weight loss (0.41 g)’. On the other hand, a sentence was added to the description of barley: ‘The grain mass loss measurements were as follows: 2.0 g for cultivar Ismena, 2.0 g for cultivar Radek, and 1.41 g for cultivar Trofeum.”

Comments 5: Please describe in more detail what specific changes were observed in the insect microbiome and metabolism in response to different grain varieties.

Response 5: Based on the conducted research, it was observed that changes in the microbiome and metabolism of insects are closely related to the type of cultivar rather than the grain species as a whole. The results indicate that the effects of individual grain cultivars differ significantly even within a single species, making it difficult to establish correlations between species. These changes are described in detail in the manuscript in subsections 2.2 and 2.3, where differences in the microbiome and metabolism of insects in response to the specific chemical properties of different cultivars are discussed.

A section was added to the discussion: The conducted research showed that changes in the microbiome, metabolism, and behavior of insects are closely dependent on the specific chemical properties of grain cultivars, rather than the grain species as a whole (Fig. 3, Fig. 5). Different cultivars within the same species uniquely affect the composition of the microbiome and metabolism of insects, which can lead to variations in feeding behaviors. For example, the analysis showed a different dominance of bacterial genera in some wheat and barley cultivars (Tab. 1). These changes influenced the metabolism of insects, suggesting that interactions between diet and microbiome have a direct impact on their development and adaptation. Furthermore, different grain cultivars can affect insect behavior, for instance, by reducing feeding efficiency through microbiome reorganization, which disrupts insect functioning and decreases their survival abilities (527-537).”

Comments 6: Although the ANOSIM results did not show significant correlations at the species level, it is worth explaining why these results are important and what implications they have for further research.

Response 6: Based on the conducted studies, it was observed that there is no correlation between grain species. This is due to the varying effects of cultivars within a specific species. To explain this in detail, this result is caused by the fact that the effect of individual samples from one cultivar was somewhat similar to some samples from another cultivar, but most samples were distinct. Therefore, the correlation and ANOVA of the main factors were insignificant because the standard error and standard deviation were slightly similar. Moreover, the result for “cultivars” is significant (both for ANOSIM and for certain chemical and developmental traits), indicating that cultivars within a species differ quite noticeably. This is further confirmed by the structural equation plots in both the manuscript and the supplement. As stated in the discussion and conclusion, patterns can only be observed within a given grain species, but there are such drastic differences between species that finding patterns is very difficult, if not impossible, unless individual traits are selected and cultivar breeding is used to regulate fatty acid levels and significantly increase antioxidant levels (although conflicting results were obtained in our study, suggesting that instead of antioxidants, future studies should include plant factors that determine ROS in insects and symbiotic microorganisms).

As for directly naming specific cultivars—this is not the purpose of this study. The aim is to indicate specific traits and relationships rather than individual cultivars. As previously mentioned, in future scientific publications, a global consortium involving multiple research teams should be established to create a worldwide grain bank and test a large number of diverse objects in terms of fatty acids, antioxidants (AOX), and reactive oxygen species (ROS), along with FTIR/Raman analysis to develop a quick and simple method for detecting traits for future cultivar development. Regarding the microbiome, the complete eradication of symbionts beneficial to the insect is still too challenging, thus the approach outlined in the letter to the editor, involving the use of natural antibiotics or bacteriostatics and the biocontrol of insects through microbiome disruption, should be developed.

A section was added to the conclusions: The lack of significant correlations at the species level, as shown by ANOSIM results, is crucial because it indicates substantial differences between cultivars within each species, which complicates the identification of general patterns between species. These results emphasize that research should focus on specific cultivars and their unique traits rather than general differences between grain species. This approach will provide a better understanding of how specific chemical properties of cultivars affect the microbiome, metabolism, and functioning of insects. The results for cultivars were significant, highlighting notable differences in microbiome structure and insect metabolism, which were also confirmed by structural equation plots. This finding has important implications for future research, suggesting that identifying specific cultivar traits may be more valuable than species-level studies.”

Comments 7: Please explain in more detail why the microbiome is so important, giving examples or references to the literature.

Response 7: Thank you for your comment. We appreciate the suggestion; however, we would like to point out that repeating detailed information about the microbiome could complicate the manuscript's readability and make it more difficult for readers to absorb the information. Our research is part of a broader cycle in which we have previously described the effects of sterilizing the symbiotic microbiome (https://doi.org/10.3390/app13031576) and the impact of various cultivars on the lesser grain borer and rice weevil. These studies were preliminary stages that set the direction for further research. The impact of the microbiome on insect functioning is briefly discussed in the introduction, lines 47 to 66, where the role of the microbiome is outlined. The aim of the current work is to highlight the key findings and indicate the research potential to encourage collaboration with other research teams. For this reason, we chose to present the main conclusions without delving into details, which will be the focus of more targeted future studies. We believe that this approach to presenting the results is more effective in encouraging further research and collaboration among scientists.

Comments 8: It is worth adding what specific changes have been observed in the microbiome, metabolism and behaviour of insects.

Response 8: The changes observed in the microbiome, metabolism, and behavior of insects mainly result from interactions with specific grain cultivars rather than entire species. Differences in the levels of fatty acids and antioxidants in the cultivars affect the microbiome, which can lead to changes in digestion, stress resistance, and overall insect behavior. The studies identified that specific traits, such as the regulation of fatty acid levels and increased antioxidant levels, can significantly influence the functioning of the microbiome and the behavior of insects, although the results regarding antioxidants were inconclusive.

This comment is similar to the comment: “Please describe in more detail the specific changes observed in the insect microbiome and metabolism in response to different grain cultivars.” In response to this question, a section was added to the discussion.

Comments 9: It is worth explaining what is meant by a lack of correlation at the species level and why this is important.

Response 9: The lack of correlation at the species level is significant because it shows that the differences between cultivars within a species are substantial enough to hinder the identification of general patterns between species. This suggests that research should focus on cultivars and their specific traits rather than on general differences between grain species. This finding highlights the need for more targeted studies that involve detailed analysis of cultivars to better understand their impact on the microbiome and metabolism of insects.

A similar comment appeared in question 6: “Although ANOSIM results did not show significant correlations at the species level, it is worth explaining why these results are important and what implications they have for further research.” In response to this question, a section was added to the text that explains the significance of these results and their implications for future research, emphasizing the need to focus on analysis at the cultivar level.

Comments 10: Please explain why the relationships between C18:2 fatty acid, entomopathogenic bacteria, impaired nitrogen cycle, lysine production of bacterial origin, and insect feeding are important and how they affect insect behaviour.

Response 10: The analysis of lysine was not the main objective of this study but was derived from data available in the scientific literature and the metabolic database MACADAM (MetaCyc+Faprotax) used in this research. We aimed to indicate that there is a connection between lysine, relevant bacterial groups, and the nitrogen cycle, and that these findings could be used to develop biomarkers for insect development. As highlighted in the literature, without lysine, development and the emergence of new generations are hindered, and the disruption of homeostasis and behavior gradually impairs the immune system, paving the way for the growth of entomopathogens. The nitrogen cycle is crucial for the metabolism of the gastrointestinal tract; symbionts reduce the concentration of the insect's classical metabolite: ammonium nitrogen and potentially ammonia, which are toxic at high concentrations. Since the share of entomopathogens has increased, they have displaced the symbionts that produce lysine. To answer how this happens and what the sequence of events is in this complex system, comprehensive analyses of metatranscriptomics and metabolomics of both the insect and microorganisms are necessary, making this an important topic for future studies.

A section was added to the conclusions: “Rearrangements in the microbiome structure contributed to an increase in the share of entomopathogens, which led to the displacement of lysine-producing symbionts. To answer how this happens and what the sequence of events is in this complex system, comprehensive analyses of metatranscriptomics and metabolomics of both the insect and microorganisms are necessary.”

Comments 11: It is worth adding what specific changes in agricultural practices may result from these findings and what agricultural and environmental benefits may result from breeding resistant cereals.

Response 11: The following section has been added to the conclusion section: “Growing cultivars of selected cereal species with specific chemical properties can significantly change agricultural practices. First and foremost, selecting and cultivating cereal cultivars that are naturally less attractive to storage pests, such as Rhyzopertha dominica, can reduce the need for synthetic pesticides, reducing storage losses and contributing to more sustainable agricultural production. This can also have a direct environmental impact, reducing soil and water pollution levels, as well as reducing negative impacts on beneficial organisms. It should also be emphasised that the use of resistant cereal cultivars will support the development of organic farming. In addition, it can increase farm efficiency by reducing the costs associated with plant protection and increasing the stability of the yields obtained.”

Comments 12: Overall, the article makes a valuable contribution to the understanding of the interaction between grain chemical properties and the storage pest microbiome, which may lead to the development of more sustainable methods to control these storage pests.

Response 12: Thank you for your positive feedback and recognition of our work. We are glad that you find our study valuable in enhancing the understanding of the interaction between grain chemical properties and the microbiome of storage pests. Our aim is indeed to contribute to the development of more sustainable pest control methods, and we appreciate your acknowledgment of the potential impact of our research in this field. Your encouraging comments motivate us to continue exploring these important interactions for future advancements in sustainable agriculture.

  1. Response to Comments on the Quality of English Language

Point 1: peptone wateR -> peptone water

Response 1: Corrected

Point 2: he genetic material -> The genetic material

Response 2: Corrected

Point 3: OUT table -> OTU table

Response 3: Corrected

Point 4: resulting in more intensive feeding and more intense development -> resulting in more intensive feeding and development

Response 4: Corrected

Point 5: Wheat cultivars such as Varius, characterised by high protein content, were preferred by the R. dominica -> Wheat cultivars such as Varius, characterised by high protein content, were preferred by R. dominica

Response 5: Corrected

Point 6: suggesting that a low-nitrogen diet may lead to greater involvement of bacterial symbionts in urea recycling processes and production of amino acids necessary for insect growth -> suggesting that a low-nitrogen diet may increase the involvement of bacterial symbionts in urea recycling processes and the production of amino acids necessary for insect growth

Response 6: Corrected
